# Plastic Softening Induced by High-Frequency Vibrations Accompanying Uniaxial Tension in Aluminum

**DOI:** 10.3390/nano12071239

**Published:** 2022-04-06

**Authors:** Ziyu Zhao, Jinxing Liu, Amir Siddiq

**Affiliations:** 1Faculty of Civil Engineering and Mechanics, Jiangsu University, Zhenjiang 212013, China; njwxchaojinchun@163.com; 2School of Engineering, University of Aberdeen, Aberdeen AB24 3FX, UK

**Keywords:** high-frequency vibrations, lattice orientation, strain rate sensitivity, preset notch, softening, molecular dynamics simulation

## Abstract

We have investigated the influences of high-frequency vibration (HFV) superimposed onto the monotonic uniaxial tension in single-crystal aluminum (Al) specimens by molecular dynamics simulations. It was found that HFV induces softening, i.e., reduction in peak stress. Similar to previous experimental results, the softening increases with the increasing HFV amplitude. Dependences on lattice orientation, tensile strain rate, and a preset notch are considered. Lattice orientation plays an important role in peak stress and plasticity. The evolution of the atomic structure reveals that dislocations have enough time to annihilate under a lower tensile strain rate, resulting in strong ups and downs in the strain–stress curves. Under a higher strain rate, newly appearing dislocations interact with previous ones before the latter annihilate, densifying the dislocation network. As a result, further dislocation motions and annihilations are considerably impeded, leading to a relatively smooth flow stage. Furthermore, by modifying the propagation direction of shear bands, a preset notch can strengthen the peak tensile stress under low-level amplitude HFVs.

## 1. Introduction

The investigation of ultrasonic treatment effects on the deformation process has continued for decades [1,2,3,4,5,6,7]. Blaha and Langenecker [8,9] studied and reported a dramatically softening (reduction in yield strength) in metal tensile or compressive tests combined with superimposed ultrasonic energy. Additionally, subsequent experimental reports [10,11,12] confirmed that the softening occurs in various metals (aluminum (Al), zinc, etc.).

To quantitively analyze the effect of ultrasonic vibrations on monotonic deformation, Yao et al. [13] proposed a modeling framework for acoustic plasticity. They modeled the ultrasonic softening and residual hardening effects based on the thermal activation theory and dislocation evolution theory, respectively. They also reported that the stress decrease owing to ultrasonic treatment is proportional to the vibration amplitude. Siddiq et al. [14,15] developed a phenomenological crystal plasticity model and a fully variational porous metal plasticity model to investigate ultrasonic softening effects. Their model can capture the ultrasonic softening effect and exhibited a trend that the plastic deformation would with ultrasonic vibrations intensity increasing. Furthermore, they also claimed that the softening phenomenon is independent of the vibration frequency in the range of 15~80 kHz. Djavanroodi et al. [16] compared conventional and ultrasonic equal channel angular pressing (ECAP) and obtained a 13% lower average stress due to the application of ultrasonic vibrations with 2.5 μm at 20 kHz. Based on finite element analysis, they also found that increasing ultrasonic vibrations amplitude and frequency during ECAP would reduce forming force simultaneously. Avik Samanta et al. [17] established a molecular dynamics model which considered the transverse ultrasonic vibrations effect to study the evolution of the diffusion layer. They have also presented a numerical approach that can predict the diffusion layer thickness during the ultrasonic welding process. Kim et al. [18] implemented the electron backscattering diffraction (EBSD) technique to observe the evolution of microstructure and mechanical properties of twinning-induced plasticity (TWIP) steels undergoing the ultrasonic vibration nano-crystalline surface modification process. Additionally, they claimed that the strength and plasticity of the material are improved by impact force. In general, previous research about the effect of ultrasonic vibrations can be summarized as follows: (i) yield strength reduces as soon as ultrasonic vibrations are imposed [11,15,19]; (ii) the magnitude of softening is found to be “proportional” to the applied ultrasonic energy intensity, i.e., the power per unit area of ultrasound [10,15,20], whereas the stress is even brought down to zero with a sufficiently high-intensity ultrasound [21]; (iii) the residual ultrasonic effect could occur even if ultrasonic vibrations stop [22,23]. It is notable that, so far, transverse vibrations have been investigated instead of longitudinal waves. The mechanism for two kinds of vibrations is unique, i.e., vibration-induced extra plastic deformation. According to plastic flow rules such as J2 plasticity, it is shear deformation that directly leads to plastic flow. In that sense, the wide adoption of transverse vibrations may be due to its relatively higher efficiency in causing plastic flow.

However, little literature is focused on the microscopic behaviors of nano Al subjected to monotonic tension accompanied by ultrasonic vibrations, and it is difficult to directly obtain microstructure evolution in real experiments. In order to reveal the underlying deformation mechanism induced by ultrasonic vibrations at the atomic level, the molecular dynamics (MD) simulation method is adopted.

The MD simulation is a good auxiliary research tool to investigate the evolution of atomic arrangements and dislocation configurations at atomic scale [24,25,26,27,28]. It can overcome the limitations of experimental conditions and exhibit atoms motion trajectories. Furthermore, the MD method can also consider a variety of factors such as strain rate [29,30,31] and lattice orientation [32,33,34]. However, MD simulation methods admittedly have an unavoidable issue than the loading rate in MD simulation is some orders of magnitude higher than in experiments. Such a fundamental issue with MD methods falls out of the scope of this study. We continue adopting MD simulations for a better understanding of undergoing atomic-level mechanisms of plastic deformation, instead of pursuing a strict correspondence to real-world experiments, as most MD-based investigations have done in the literature [35,36,37,38]. Thus, to investigate the softening phenomenon on the tensile properties of nano-scaled single crystal Al, we impose high-frequency vibrations (HFV) to mimic ultrasonic vibrations in the MD simulation method. Indeed, we continue to mimic ultrasonic vibrations, as HFV in such MD simulations do show some features such as plastic softening observed experimentally.

In the present study, we use MD simulation methods to examine the microscope behaviors and associated deformation mechanisms of nano-Al under a monotonic tension accompanied by HFV, with features such as lattice orientation, tensile strain rate, and preset notch taken into account.

## 2. Materials and Methods

As shown in Figure 1, we employ the atomic simulator LAMMPS [39] to investigate the deformation behavior in nano-scaled single crystal Al subjected to uniaxial tension combined with HFV. The specimens have a size of 12.15 nm × 17.21 nm × 17.21 nm, containing 21600 atoms. This size is sufficient to study atomic structure evolution and the dislocation nucleation process [40,41]. Periodic boundary conditions are applied in all three directions. The lattice orientations of the specimens are (1) X-[100] Y-[011] Z-[0-11], (2) X-[100] Y-[010] Z-[001], and (3) X-[11-1] Y-[112] Z-[1-10]; in the rest of this study, these orientations will be referred as O#1, O#2, and O#3, respectively.

We adopt the velocity Verlet algorithm to integrate the equation of motion, with a time step of 1 fs [39]. A Nosé–Hoover thermostat and barostat were chosen to control the temperature and pressure [42]. The temperature of the system was set at 10 K during the whole MD simulation process, which can avoid the effect of temperature [36,43]. The pressure along the x- and y-directions were maintained at 0 bar by applying the constant particle number/pressure/temperature (NPT) ensemble. The time of relaxation was 200 ps, followed by a tensile strain along with the z-axis, and HFVs with various amplitude (A = 1~5 Å) formed as the cyclic shear load was applied on the top layer atoms (blue atoms in Figure 1a); the frequency of vibration was 10^12^ Hz. Generally, under a fixed frequency, a higher vibration amplitude corresponds to a larger power flux density and therefore a higher degree of plastic deformation. The strain rate of the tensile load ranges from 10^8^ to 10^10^ s^−1^. The duration of tension has a magnitude of time 100 ps. In addition, specimens with one notch in the center that run in the Y-direction were also investigated, the cross-sectional dimension of the notch is 5 Å × 10 Å, and the sample is shown in Figure 1b. In this article, the axial stress along the tensile direction *P*_zz_, is calculated based on the virial formula [44], i.e.,
(1)$$P_{zz}=\frac{1}{V}\sum_{i=1}^{N}\left(m_{i}v_{iz}^{2}+r_{iz}f_{iz}\right)$$
where *V* is the ensemble volume, *m_i_* and *v_iz_* are, respectively, the mass and the axial velocity of the *i*th atom, *r_iz_* and *f_iz_* are the atom position vector and the axial component of the resultant force on the *i*th atom.

For the studied single crystal Al, we adopt an embedded-atom method (EAM) potential parameters supplied by Mendelev et al. [45], which has been widely used [46,47,48,49,50] to estimate the inter-atomic interactions of atoms Al in specimens. The potential energy of a material system is given by
(2)$$E_{i}=F_{\alpha }\left(\sum_{j\neq i}\rho_{\alpha }{}_{\beta }\left(\gamma_{ij}\right)\right)+\frac{1}{2}\phi_{\alpha \beta }(\gamma_{ij})$$
where *F* is the embedding energy which is a function of the atomic electron density *ρ, ϕ* is a pair potential interaction, and *α* and *β* are the element types of atoms *i* and *j*, respectively. The multi-body nature of the EAM potential is a result of the embedding energy term. We use OVITO [51] as the visualization tool to show the trajectories of atoms during the deformation process. Common Neighbor Analysis (CNA) [52,53] and the Dislocation Extraction Algorithm (DXA) [54] based on OVITO are used to analyze atomic structures and dislocations, respectively. We have confirmed that, for a fixed specimen, the results of various runs always come to a very steady convergence. Thus, in this context, all diagrams are plotted based on single simulation runs. Notably, the settings adopted in the present study are still challenging in practical experimental designs. Nevertheless, it does not influence the present simulations to provide us a better understanding of deforming mechanisms in the tension process accompanied by HFV.

## 3. Results

This section may be divided into subheadings. It should provide a concise and precise description of the experimental results, their interpretation, and the experimental conclusions that can be drawn.

### 3.1. Lattice Orientation Effect

We employ the loading and boundary conditions in Figure 1, and simulate three kinds of lattice orientations, i.e., O#1: x-[100] y-[011] z-[0-11], O#2: x-[11-1] y-[112] z-[1-10], O#3: x-[100] y-[010] z-[001]. The loading tensile strain rate is set as 10^9^ s^−1^ for all. For each kind of orientation, six cases are investigated, i.e., the amplitude of HFVs of frequency 10^12^ Hz is 0, 1, 2, 3, 4, and 5 Å, respectively. The stress–strain responses captured after subjecting single crystal Al to different lattice orientations are plotted in Figure 2. The results suggest a strong dependence on lattice orientations. For all three orientations, it can be shown that the peak stress, which is defined as the maximum stress on the stress–strain curve, drops as the HFV amplitude increases. This softening trend coincides qualitatively with experimental observations [15]. Correspondingly, the elastic-deforming span decreases as HFV amplitude increases. Here, O#3 has the strongest amplitude-related softening, whereas O#1 has the weakest. This is explained from the viewpoint of the Schmid factor. Under three orientations, the tension is applied along the [0-11], [1-10], and [001] crystallographic orientations, respectively. For O#1, 4 slip systems with the highest Schmid factor (~0.40824) are found to be activated simultaneously: (10-1) [1-11], (110) [1-11], (1-10) [11-1] and (101) [11-1]. For O#2, 4 slip systems with the highest Schmid factor (~0.40824) are found to be activated simultaneously: (10-1) [1-11], (10-1) [111], (01–1) [-111] and (101) [-111]. For O#3, 8 slip systems including (101) [-111], (101) [11-1], (110) [-111], (110) [1-11], (10-1) [111], (10-1) [1-11], (-110) [111], and (-110) [11-1] have the highest Schmid factor (~0.40824).

It can be observed that the decline in stress is negligible for O#1 while the amplitude increases from 0 to 1 Å (Figure 2). When the amplitude reaches 2 Å, an obvious softening phenomenon appears in O#3. Meanwhile, the decrease in PS is still inapparent for the case of O#1 and 2. After that, as the HFV amplitude continues to grow, the peak stress of O#2 drops dramatically while amplitude rises by more than 3 Å. However, the peak stress of O#1 still decreases smoothly. It suggests that the softening phenomenon is most remarkable in O#3 and the difference of peak stress between the case with A = 0 and 5 Å is 7.428 GPa, whereas it is only 2.13 GPa for O#1. Again, such simulations on softening should be considered to be qualitatively helpful for better understanding the atomic-level mechanism, regarding the differences between simulating and experimental loading rate settings. Additionally, it can also be observed from O#3 that all cases with HFV fail after peak stress. This indicates that the plasticity of O#3 models are highly weakened. In addition, the stress response of O#1 at room temperature is shown in Figure 2d. It can be seen that the peak stress also decreases with amplitude increasing, which is similar to cases at 10 K. Of course, the absolute values of peak flow stresses decrease with temperature. The temperature dependence is believed to be another interesting topic, which, however, has fallen out of the scope of the present focus. For brevity, the following analyses are all at 10 K.

To gain a better understanding of the deformation regulating mechanism, the deformed constructions of O#1 and 3 are subjected to dislocation extraction algorithm (DXA) and common neighbor analysis (CNA), and the results are shown in Figure 3 and Figure 4, respectively. In the atomic configuration, green atoms are recognized as face centered cubic (FCC) represent perfect Al crystal, red atoms are recognized as hexagonal closest packed (HCP) represent stacking faults (SFs), which is induced by slips of 1/6<112> Shockley partial dislocations, and white atoms are recognized as ‘other’ are those with no crystal symmetry, and mainly include amorphous atoms produced by extremely strong HFVs. In dislocation configurations, the green, blue, and red lines represent Shockley partial type dislocations, perfect type dislocations, and other type dislocations, respectively. The evolution of dislocation total length for each model is shown in Figure 5.

Figure 3a–c show the evolution of atomic configuration of specimens with lattice O#1 with A = 0, 1, and 3 Å. As can be seen, the deformation motion is greatly affected by HFV. For the monotonic tensile test, a higher fraction of dislocation and SFs can be observed during the process. However, the application of HFV significantly alters the propagation of SFs and dislocations slip. For A = 1 Å, plenty of SFs in the same direction appear in the specimens with a low-density dislocation. This indicates that HFVs caused dislocations to slip through the specimen. For A = 3 Å, little dislocation and almost no SF exist in the specimen at strain equal to 0.125 and 0.171, revealing that the specimen’s deformation mode is changed by high-amplitude HFVs.

Compared with the results for lattice O#1, i.e., Figure 2 and Figure 3, O#3 show some new characteristics. In HFV testing, there is essentially no plastic deformation, and the specimens are quickly split into two by a macro-crack. The fracture processes correspond to the serrate processes following peak stress for A = 1~5 Å in Figure 2c. As seen in Figure 4, HFV vibrations alter the deformation mechanism when compared to monotonic stress. Only a finite amount of perfect and Shockley partial type dislocations can be seen in the neighborhood of the crack surfaces, especially for high amplitude. Additionally, almost no SF appears in the specimens. This indicates that, for such a lattice orientation, it is very hard for any dislocation to slip; therefore, the specimen behaves like a brittle medium. This means that the influence for HFV becomes equal to and even dominant over the monotonic tension.

The evolutions of total dislocation length are shown in Figure 5. It can be found that the emission of dislocation is substantially impacted by the HFV. A larger amplitude leads to an earlier onset of dislocation motions indicating a sharp peak in dislocation length. This trend is essentially the same thing as that of peak stress. The dislocation density decreases with amplitude increasing, which is also similar to the trend of peak stress. During crystalline plastic deformation, a larger spatial dispersion of dislocations leads to the production of more SFs. Stacking defects with low formation energies restrict dislocation motion, resulting in material strengthening. The maximum dislocation density corresponds to the specimen without HFV, which has the highest peak stress and the minimum dislocation density, while the specimen with 5 Å amplitude HFV, which has the lowest peak stress in Figure 2a.

#### The Effect on Plasticity

The plasticity of materials is a topic that is frequently researched. In this part, the effect of the HFV on specimen plasticity is explored. HFV specimens fail shortly at peak stress in O#3. However, on the other hand, different results are shown in O#1 and O#2. The fracture strain versus amplitude of O#1 and O#2 are shown in Figure 6. As expected, the fracture strain increases significantly with increasing HFV amplitude under low amplitude (A = 1 and 2 Å). Hereafter, the effect on plasticity has changed when amplitude continuously increases (Amplitude ≥ 3 Å). For O#1, the increase in fracture strain becomes insignificant. However, for O#2, an inferior ductility is shown with amplitude increasing. The fracture strain test with A = 3 Å is already lower than monotonic tension. Thus, the results indicate that the influence of HFVs on elongation depends on its amplitude. A superior elongation can be obtained in the test with HFVs with amplitude ranging from 1 Å to 2 Å. When the amplitude reaches a larger value (>3 Å), the effect of HFV on specimen elongation shifts from enhanced to weakened.

To further explore the effect of HFV on plasticity, we take, for example, the specimen with lattice O#2 under HFVs amplitudes A = 0 and 1 Å, respectively, as shown in Figure 7. As can be seen, the fracture mode of two separate specimens is the same. Much fewer SFs remain, corresponding to a much lower probability of dislocation at breaking strain. In the case of A = 1 Å, however, there are many more shear strain concentration zones in the specimen, and they are distributed uniformly across the entire range. This suggests that appropriate amplitude HFVs can remarkably promote the deformation non-localization, resulting in the enhancement of the elongation of the specimen.

### 3.2. Strain Rate Effect

We investigate the influence of tensile strain rate on the behaviors of the specimen with HFV frequency fixed. To this end, three rates are studied, i.e., 10^8^, 10^9^, and 10^10^ s^−1^ for the specimen with lattice O#1.

The stress responses are dependent on the prescribed strain rate, as shown in Figure 2 and Figure 8. First, only a negligible difference is seen in the pre-yield regime under various rates. The higher the applied tensile rate, the weaker dependence YS has on the HFV amplitude. The most obvious difference exists in the post-peak regime. For the highest rate, i.e., 10^10^ s^−1^ (Figure 8b), the curves following peak stress are the smoothest and flattest at a roughly equal stress level, whereas for the lowest rate, i.e., 10^8^ s^−1^ (Figure 8a), the post-peak curves become zigzagged. For the mediate rate (Figure 2), the zigzagged feature is also observable and becomes weaker than in Figure 8a.

The above trends can be explained by analyzing the dislocation motions in specimens, as shown in Figure 9. For the lowest rate, i.e., Figure 9a, when the strain is 8.32%, dislocations are emitted, and SFs appear simultaneously. However, as strain reaches 10.43%, there are no stored dislocations but only stacking fault planes running across the whole specimen, which corresponds to the second peek of the red curve in Figure 8a. It means that dislocation annihilation occurs most easily due to the lowest tensile rate because dislocation lines have enough time to be driven out of the specimen by HFV. Afterward, as strain reaches 11.74%, due to reaching another critical amount of elastic energy stored, dislocations are re-activated in the specimen, which means the appearance of a second yield. Then, the subsequent dislocations will again be driven out as before, and so on. In contrast, for the highest strain rate, the dislocation network keeps becoming tenser and tenser (Figure 9b). The initial emission of dislocation occurs at strain = 8.93% and SFs appear simultaneously. Dislocations and SFs keep increasing with tensile strain increasing. Under such a high rate, subsequent dislocations appear before those already existing slip out of the specimen under the drive of HFV; thus, inter-crossings of dislocations keep happening and serve as a significant mechanism of arresting active dislocations. This leads to the continuous accumulation of dislocations and SFs.

### 3.3. Notch Effect

In this subsection, the effect of a notch on the deformation mechanism is considered. Crystal O#1 (X-[100] Y-[011] Z-[0-11]) is used for simulations, and a notch is pre-aligned in the X-direction. The strain rate is set as 10^9^ s^−1^. The stress–strain responses for different amplitudes are plotted in Figure 10. Unlike previous trends, the largest peak stress is seen at A = 1 Å. The peak stress of the monotonic test is 4.17 GPa, whereas A = 1 Å has a peak stress of 4.92 GPa. Hereafter, the peak stress drops quickly as amplitude rise to 3 Å.

To have a better insight into the deformation mechanism, the atomistic configuration and shear strain distributions of specimens at peak stress with various amplitude HFVs are shown in Figure 11. It is easy to see that the propagation of shear concentration bands in circumstances of varying amplitude is quite diverse. For the monotonic case, shear bands are activated from the preset notch. Then, they propagate along with slip systems. On the other hand, shear bands in the case with A = 3 Å parallel with each other are induced by the high amplitude HFV, which means HFV plays an important role in the deformation process. Interestingly, for A = 1 Å, the propagation mode of shear concentration bands is a mixture of the above two, which shows that the influence of tension and HFVs, in this case, are in the same order. The interlaced shear bands greatly increase the peak stress of the specimen.

## 4. Conclusions

In the present study, we have examined deformation behaviors and mechanisms in nano-scaled single crystal Al subjected to uniaxial tension combined with HFVs by MD simulations. HFVs induce plastic softening because they accelerate the activation of dislocations and shorten the elastic stage. The softening in our study is similar to that of previous experimental results. The main conclusions are summarized as follows:Different lattice orientations can have different Schmid factors and result in various extents of softening. The biggest softening is shown in O#3, and the extent of softening increases with HFV amplitude. Lattice orientation also highly affects plasticity. On one hand, the plasticity of the specimen with O#1 and O#2 is enhanced by low-level amplitude HFV. On the other hand, owing to the lack of plastic stage, the failure of O#3 occurs rapidly with all level amplitude HFVs.The extent of softening increases with the tensile strain rate decreasing. A lower strain rate permits dislocations to have enough time to slip out of the specimen, and it results in specimen yields many times. In contrast, a higher strain rate would weaken the softening phenomenon. It leads to the formation of dense dislocation networks by rapid and continuous dislocations activations, and the networks impede dislocations slip and annihilation.For O#1, a preset notch produces a distinct trend on stress response from free-defect specimens. The propagate motion of shear bands for monotonic tensile test and high amplitude (≥3 Å) HFVs are inconsistent. In the case of A = 1 Å, however, two separate propagation motions of shear bands are blended, resulting in strength enhancement.

## Figures and Tables

**Figure 1 nanomaterials-12-01239-f001:**
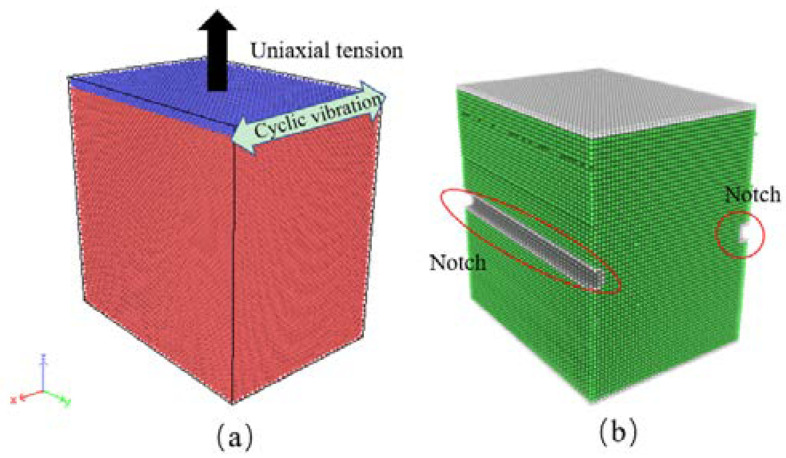
Schematic diagrams of (**a**) tension and HFV test on Al specimen, (**b**) specimen with one notch.

**Figure 2 nanomaterials-12-01239-f002:**
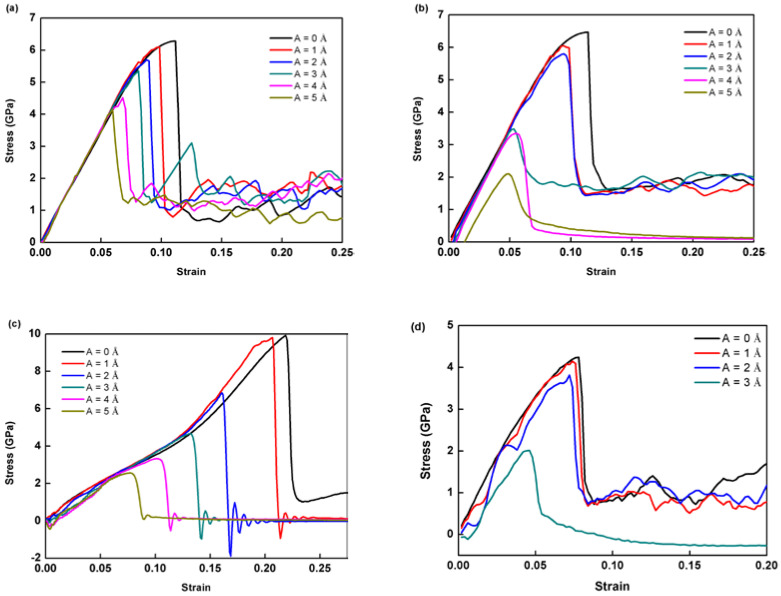
Stress responses in single crystal Al specimens according to crystal: (**a**) O#1 (x-[100] y-[011] z-[0-11]), (**b**) O#2 (x-[11-1] y-[112] z-[1-10]), (**c**) O#3 (x-[100] y-[010] z-[001]) subjected to uniaxial tension accompanied by HFVs of various amplitudes, and (**d**) O#1 at room temperature (300 K).

**Figure 3 nanomaterials-12-01239-f003:**
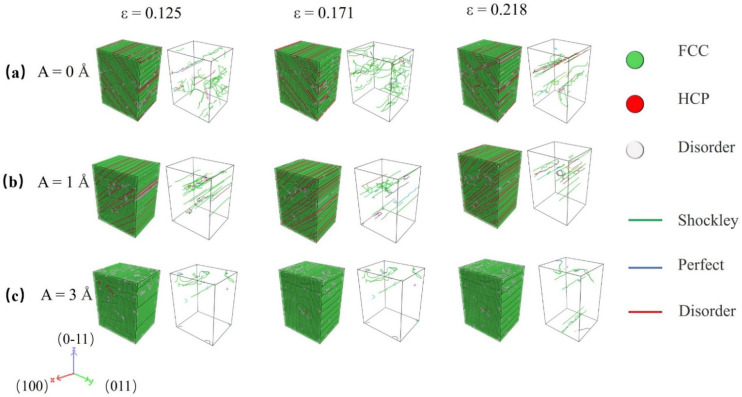
Evolutions of atomic and dislocation structure in specimens with O#1 under various HFV amplitudes: (**a**) A = 0 Å, (**b**) A = 1 Å, and (**c**) A = 3 Å.

**Figure 4 nanomaterials-12-01239-f004:**
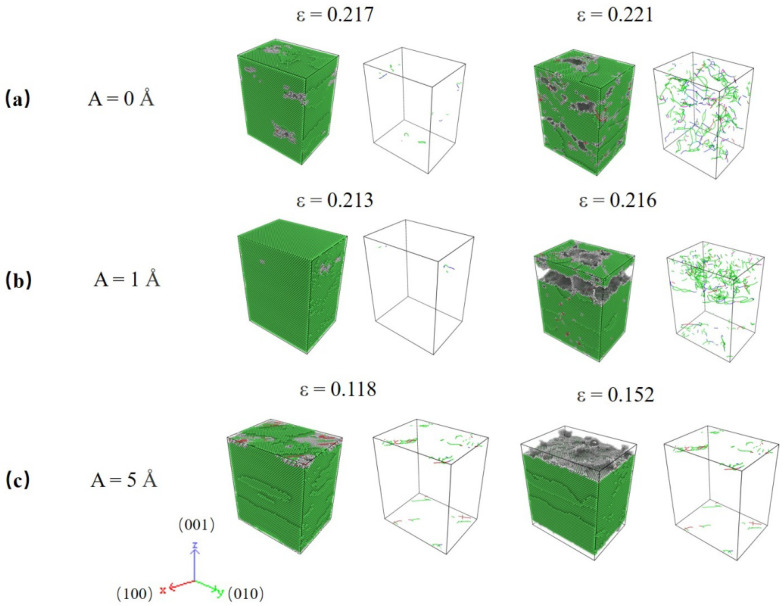
Atomic and dislocation structure for specimens under varied HFV amplitudes at peak stress strain: (**a**) A = 0 Å, (**b**) A = 1 Å, (**c**) A = 5 Å. The lattice orientation of specimens is X-[100] Y-[010] Z-[001].

**Figure 5 nanomaterials-12-01239-f005:**
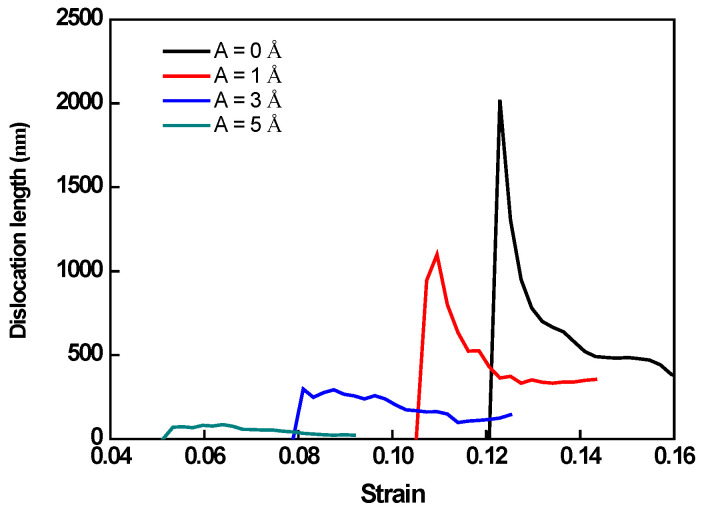
Evolutions of total dislocation lengths in specimens according to crystal O#1 under HFV amplitudes A = 0 Å, 1 Å, 3 Å, and 5 Å, respectively.

**Figure 6 nanomaterials-12-01239-f006:**
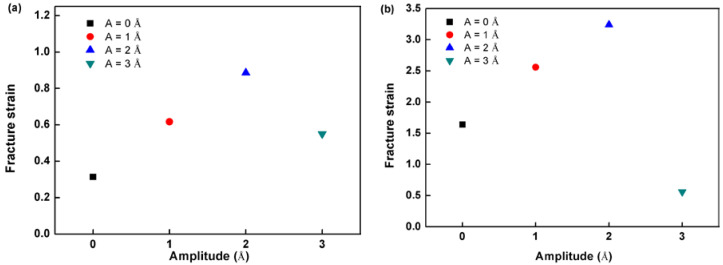
Fracture strain versus HFV amplitude according to crystal (**a**) O#1 and (**b**) O#2.

**Figure 7 nanomaterials-12-01239-f007:**
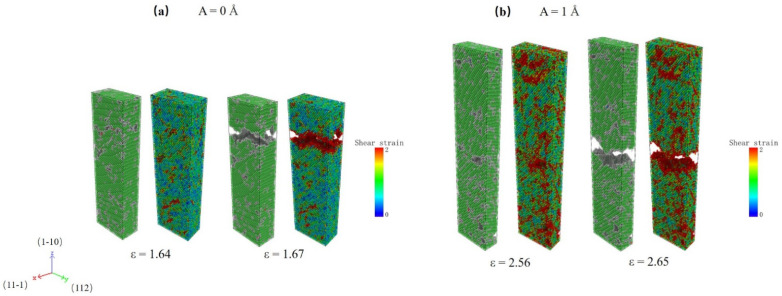
Atomic configuration and shear strain distribution of specimens according to crystal O#2 at fracture strain with (**a**) A = 0 Å and (**b**) A = 2 Å.

**Figure 8 nanomaterials-12-01239-f008:**
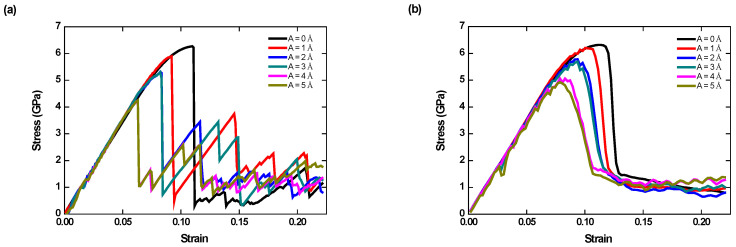
Stress–strain responses for the specimens according to crystal O#1 and various HFV amplitudes under the tensile strain rates (**a**) 10^8^ s^−1^ and (**b**) tensile strain rate =10^10^ s^−1^.

**Figure 9 nanomaterials-12-01239-f009:**
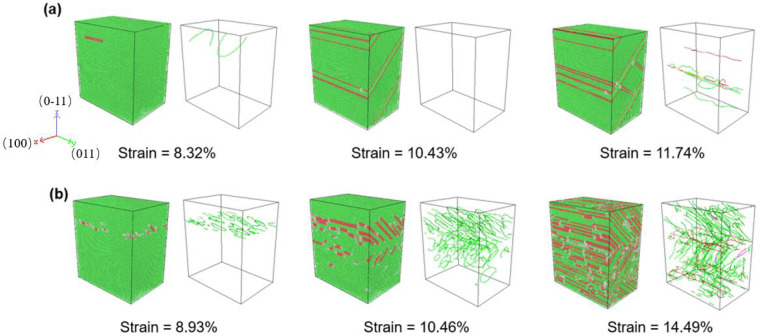
Evolution of atomic and dislocation configurations for specimen (A = 2 Å) with (**a**) tensile strain rate = 10^8^ s^−1^, (**b**) tensile strain rate = 10^10^ s^−1^.

**Figure 10 nanomaterials-12-01239-f010:**
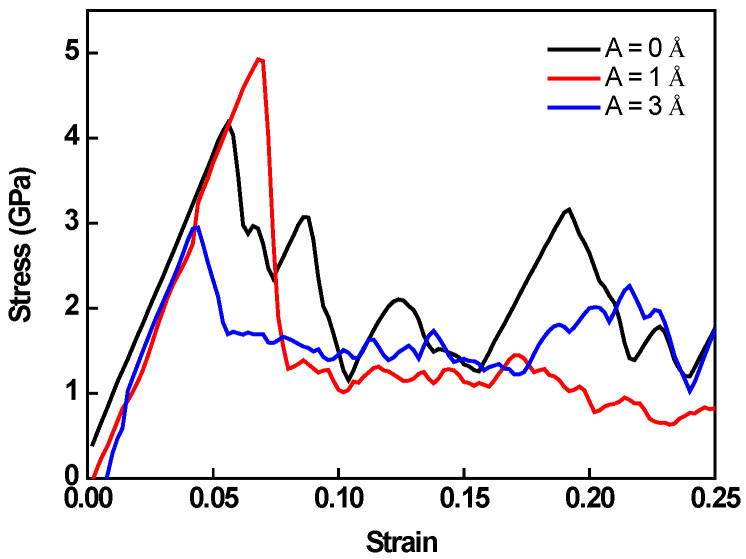
Stress–strain response for the specimens with a preset notch according to crystal O#1.

**Figure 11 nanomaterials-12-01239-f011:**
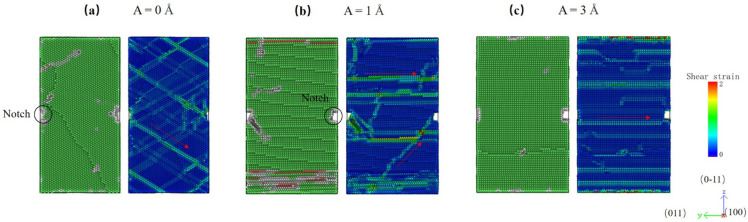
Atomic configuration and shear strain distribution of specimens with a preset notch under HFVs at peak stress, (**a**) A = 0 Å, (**b**) A = 1 Å, and (**c**) A = 3 Å.

## Data Availability

The data that support the findings of this study are available from the corresponding author upon reasonable request.

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
