# Peer review of "Plastic Softening Induced by High-Frequency Vibrations Accompanying Uniaxial Tension in Aluminum"

_nanomaterials, 2022, doi:10.3390/nano12071239_

Round 1

Reviewer 1 Report

The authors use MD simulations to study the effects of transversal ultrasonic vibration on aluminum crystals under tension. They report that ultrasonic vibrations induce softening of the material. Lattice orientation also plays an important role.

I believe that the study is sound and potentially interesting for Nanomaterials, but some details need to be improved first.

  1. The authors chose to study transverse ultrasound waves. Do they bear practical importance (can they be used in practice)? Are longitudinal waves expected to have smaller effects on dislocations?

  1. What force field was used to model the crystal?

  1. Are the results presented in figures obtained from single simulation runs or averaged over several runs? What are the simulation times?

  1. Please clarify why the simulations were performed for the temperature of 10 K. Are the results relevant, at least qualitatively, for room temperature?

  1. Please explain better the notation for crystal orientations. Is there a way to visualize them?

  1. Can the amplitudes of the used ultrasonic waves be related to power flux density or to dB? Are these values experimentally realizable?

  1. Are there really two notches explored as stated in the text? In Fig 1b, assuming PBC, I can see only one notch.

  1. The legends in Fig. 6 seem to be redundant.

  1. English is in general good, but in some parts of the text, it can be improved. Sometimes the adjective "ultrasonic" is missing the noun, e.g., ultrasonic waves, ultrasonic vibrations, etc. Typo in: "Face cube Face centered cubic (FCC) " in line 154.

Reviewer 2 Report

  The article by Ziyu Zhao, Jinxing Liu, and Amir Siddiq aimed at a nanoscale understanding of the ultrasonic effect on the tensile properties of Al. The authors implemented the molecular dynamics (MD) simulation to investigate the evolution of atomic arrangements and dislocation configurations at atomic-scale in single-crystal Aluminium (Al) specimens. The article has interesting results, but they cause my doubts, which do not allow recommending the article for publication. My comments are below.

  The main doubt that this work raises is not the physical values of the applied strain frequencies. It is not entirely clear why the authors call the frequency 10 ^ 12 Hz superimposed to the created samples as ultrasonic vibration? Usually, which by the way is mentioned in one of the cited works (Ref. 15), ultrasonic frequency is understood as frequencies of 15–18 kHz! It is not entirely clear how such frequencies can be excited in a real experiment? Thus, the simulation conditions correspond to the non-physical frequency range, which cannot be called "the ultrasonic". Based on this, we can conclude that all the results obtained correspond to exotic conditions that are not realizable in reality. In this regard, the authors need to make a serious justification for the choice of modeling conditions. It should also be indicated how the case under study can be realized in a real experiment?

  Less significant remarks.

  The authors do simulations at 10K. Why was this temperature chosen? This should be justified. In particular, it would be good to provide references to experimental works where the deformation of aluminum was studied at similar temperatures.

  Figures 3, 5, 7, 9, and 11 should be made larger and in bigger resolution.

Round 2

Reviewer 2 Report

I agree with the authors' corrections. The article may be recommended for publication.